# Radiomics for Discrimination between Early-Stage Nasopharyngeal Carcinoma and Benign Hyperplasia with Stable Feature Selection on MRI

**DOI:** 10.3390/cancers14143433

**Published:** 2022-07-14

**Authors:** Lun M. Wong, Qi Yong H. Ai, Rongli Zhang, Frankie Mo, Ann D. King

**Affiliations:** 1Department of Imaging and Interventional Radiology, Prince of Wales Hospital, Faculty of Medicine, The Chinese University of Hong Kong, Hong Kong SAR, China; lun.m.wong@cuhk.edu.hk (L.M.W.); zhangrongli@link.cuhk.edu.hk (R.Z.); 2Department of Health Technology and Informatics, The Hong Kong Polytechnic University, Hong Kong SAR, China; 3Department of Clinical Oncology, State Key Laboratory of Translational Oncology, Sir YK Pao Centre for Cancer, Hong Kong Cancer Institute and Prince of Wales Hospital, The Chinese University of Hong Kong, Hong Kong SAR, China; frankie@clo.cuhk.edu.hk

**Keywords:** radiomics, nasopharyngeal carcinoma, benign hyperplasia, magnetic resonance imaging, feature selection stability, machine learning

## Abstract

**Simple Summary:**

In this study, we investigated the potential of radiomic models to discriminate nasopharyngeal carcinoma from benign hyperplasia on MRI, which is important to enable screening programs to detect cancer early. We found that whereas radiomics showed promising performance, instability was presented by the feature selection step in the radiomics pipeline, which could undermine its reliability. Therefore, we built a radiomics model using 17 features selected from a pool of 422 features by a proposed ensemble technique that improved the feature selection stability using a combination of bagging and boosting. This radiomic model achieved an area under the curve of 0.85 and 0.80 for discriminating the two abnormalities on the training and testing data, respectively. In addition, the proposed feature selection technique significantly improved stability when compared to well-established techniques.

**Abstract:**

Discriminating early-stage nasopharyngeal carcinoma (NPC) from benign hyperplasia (BH) on MRI is a challenging but important task for the early detection of NPC in screening programs. Radiomics models have the potential to meet this challenge, but instability in the feature selection step may reduce their reliability. Therefore, in this study, we aim to discriminate between early-stage T1 NPC and BH on MRI using radiomics and propose a method to improve the stability of the feature selection step in the radiomics pipeline. A radiomics model was trained using data from 442 patients (221 early-stage T1 NPC and 221 with BH) scanned at 3T and tested on 213 patients (99 early-stage T1 NPC and 114 BH) scanned at 1.5T. To verify the improvement in feature selection stability, we compared our proposed ensemble technique, which uses a combination of bagging and boosting (BB-RENT), with the well-established elastic net. The proposed radiomics model achieved an area under the curve of 0.85 (95% confidence interval (CI): 0.82–0.89) and 0.80 (95% CI: 0.74–0.86) in discriminating NPC and BH in the 3T training and 1.5T testing cohort, respectively, using 17 features selected from a pool of 422 features by the proposed feature selection technique. BB-RENT showed a better feature selection stability compared to the elastic net (Jaccard index = 0.39 ± 0.14 and 0.24 ± 0.06, respectively; *p* < 0.001).

## 1. Introduction

The success of nasopharyngeal carcinoma (NPC) screening programs in detecting asymptomatic individuals with early-stage NPC can considerably improve the survival and quality of life of NPC patients [1]. It has recently been shown that adding MRI to endoscopic examination can improve the detection of early-stage NPC in individuals referred for investigation following a positive plasma Epstein–Barr virus (EBV) DNA screening test [2]. However, benign hyperplasia (BH) causes problems for NPC detection using MRI. In a previously conducted NPC high-risk group screening program, we observed that BH occurred in up to 83% of EBV-DNA screen-positive patients [3]. Some of these BHs may overlap in appearance with early-stage NPCs on MRI, leading to false-positive detection of NPC (Figure 1) [4]. A proposed MRI grading system for visual inspection of contrast-enhanced images [5] was recently modified to improve the MRI performance [4]. With screening programs in mind, a second MRI grading system has also been proposed for application with non-contrast-enhanced MRI sequences [6]. This MRI grading system has the disadvantage of requiring an experienced observer and showed slightly reduced performance when compared to the system using contrast-enhanced images. Deep convolutional neural networks (CNNs) have shown promise in automating discrimination between non-contrast-enhanced scans [7,8] and contrast-enhanced scans [9], summarized in Table 1, but CNNs are often criticized for their opaqueness and lack of interpretability. Therefore, we are interested in searching for a comprehensive analytical method that can discriminate between NPC and BH, such that we could eventually develop a transparent system that can be widely applied in future NPC screening programs.

Radiomics involves high-throughput analyses of radiology features for the characterization of tissues, which offers the possibility of sparing invasive procedures and accelerating the workflow in many clinical aspects, including cancer management for NPC [10,11]. Radiomics analysis generally relies on handcrafted features that have better transparency compared to CNNs and is therefore a natural candidate with respect to our interest in discrimination of the two considered abnormalities. However, the role of radiomics in discriminating NPC from BHs has not yet been investigated. Furthermore, the rapid development of radiomics over the past decade has been recently overshadowed by increasing concerns regarding its reproducibility and reliability [12,13,14], including for head and neck MRI [15]. In particular, a key obstacle is the instability of the feature selection step, whereby even minor alterations in the training data may appreciably influence the features selected, hampering the confidence in adopting the discovered radiomics models for clinical applications. In light of this, methods to improve feature selection stability for medical imaging have rapidly gained attention. Several recent publications have highlighted the use of ensemble methods to improve the feature selection stability [16,17,18,19], mainly investigating three ensemble techniques, namely resampling, bagging, and boosting; however, their combined use has not been studied in radiomics. It is possible that using these techniques in combination could further improve the stability.

Therefore, the purpose of this study was to investigate the role of radiomics analysis in discrimination between early-stage T1 NPC and BH on MRI. We built a radiomics model for discrimination using features selected using the technique proposed in this study, which combines bagging, boosting, and the repeated elastic net technique (RENT) [17], which we call bagged-boosted RENT (BB-RENT). Using radiomic features extracted from the non-contrast-enhanced, T2-weighted, fat-suppressed (T2w-fs) MRI, the model was trained using a patient cohort scanned with a 3 T MRI scanner (3 T MRI training cohort) with fivefold cross validation and tested on a patient cohort scanned with 1.5 T MRI scanners (1.5 T MRI testing cohort).

## 2. Methods and Materials

### 2.1. Patient Characteristics

This retrospective study was approved by the local institutional ethics review board, and the requirement of written consent was waived due to the retrospective nature of the study.

Two ethically approved Chinese patient cohorts that had undergone a T2w-fs MRI scan of the nasopharynx in our institution on a 3 T MRI scanner between February 2010 and March 2021 (the 3 T MRI training cohort) and on 1.5 T MRI scanners between March 2005 and January 2010 (the 1.5 T testing MRI cohort) were included for training and testing of the radiomic model, respectively. Each cohort comprised two groups of patients:
(i)Those with new biopsy-proven, undifferentiated, or poorly differentiated NPC stage T1, according to the 8th edition of guidelines by the American Joint Committee on Cancer/Union of International Cancer Control [20,21], with a segmentable thickness of ≥3 mm on at least one axial slice; and(ii)Those with BH with a segmentable thickness of ≥3 mm on at least one axial slice without any evidence of NPC on MRI or endoscopic examination, followed-up for a minimum of 12 months.

MRI scans with non-standard spacing or with artefacts obstructing the primary tumor segmentation were excluded from the analysis. Finally, 442 patients (comprising 221 NPC and 221 BH patients) in the 3 T MRI training cohort and 213 patients (comprising 99 NPC and 114 BH patients) in the 1.5 T MRI testing cohort were included for analysis. Details of the study flow are reported in Figure 2.

### 2.2. Image Acquisition and Preprocessing

For the 3 T MRI training cohort, MRI was performed using a Philips Achieva TX 3.0-T (Phillips Healthcare, Best, The Netherlands) machine containing a body coil for radio frequency transmission and a 16-channel neurovascular phased-array coil for reception. Patients were scanned with standard sequences centered at the nasopharynx, including an axial MRI T2w-fs (repetition time/echo time = 4060/80, train length = 17, field of view = 230 × 185 mm, number of slices = 30–36, sensitivity encoding factor = 1, slice thickness = 4 mm, pixel size = 0.45 mm × 0.45 mm to 0.49 mm × 0.49 mm).

For the 1.5 T MRI testing cohort, MRI was performed using either a Phillips Intera NT 1.5 T MRI (Phillips Healthcare, Best, The Netherlands) or a Siemens 1.5 T Sonata MRI (Siemens Medical System, Erlangen, Germany), both containing a body coil for radio frequency transmission and a 16-channel receiver coil. Patients were scanned with standard sequences centered at the nasopharynx, including an axial MRI T2w-fs (repetition time/echo time = 2000/75, train length = 20, field of view = 230 mm × 185 mm, number of slices = 30, sensitivity encoding factor = 1, slice thickness = 4–5.5 mm, pixel size = 0.32 mm × 0.32 mm to 0.90 mm × 0.90 mm).

Prior to feature extraction, all of the MRI images were normalized in three steps: (i) unification of pixel sizes to 0.45 mm × 0.45 mm and slice thickness to 4 mm using linear interpolation; (ii) application of bias-field correction using the N4ITK method [22]; and (iii) normalization of the intensity profiles using Nyul normalization [23] with the tissue mask (foreground mask) generated using Huang’s method [24].

### 2.3. Lesion Delineation and Feature Extraction

The primary NPC and BH were manually delineated on each T2w-fs image, with reference to all available sequences by an expert (Q.Y.A.) with 7 years of experience in head and neck MRI. To evaluate the intraobserver variation of the delineation and its effect on the stability of the extracted features, a second set of delineations of the abnormalities in the nasopharynx was performed by the same expert after a time interval of at least 14 days.

A total of 422 radiomic features were extracted from each of the T2w-fs images using a well-established software package, PyRadiomics (Version 3.0.1, Harvard Medical School, UK) [25]. We excluded features derived from wavelet filters in this study, as this group of features is not rotationally invariant and involves an excessive number of features that could increase the risk of overfitting. A detailed list of the extracted features and the configurations for feature extraction is provided in the (Appendix A). Two sets of the same features were extracted, corresponding to the two sets of manual delineation, for intraobserver variation analysis.

### 2.4. Feature Selection

To reduce the model complexity and control the risk of overfitting, feature selection was performed in two steps. The first step preliminarily filtered the features using statistical properties, whereas the second step selected from the remaining features using the proposed BB-RENT. The details of these two steps are provided in the following sections.

#### 2.4.1. Preliminary Feature Filtering

Prior to supervised feature selection, features were preliminarily filtered using quantitative criteria designed with the idea that (i) useful features should be robust enough to tolerate reasonable variations in the manual delineation and that (ii) they should have a significant association with the pathology of interest.

For the first criterion, the intraclass correlations (ICCs) of each radiomic feature obtained from the two sets of manual delineations were computed, and any radiomic feature with an ICC less than 0.9 or a *p*-value greater than 0.05 were excluded [26]. For the second criterion, Student’s *t*-test or non-parametric Mann–Whitney *U*-test—depending on the normality of the data—were used to identify features that can discriminate between NPC and benign hyperplasia. Any feature with a *p*-value greater than 0.05, suggesting an insignificant correlation, was excluded. In addition, to avoid arithmetic errors when computing these two criteria, a variance threshold was used to remove all-zero or all-one features.

#### 2.4.2. Supervised Feature Selection

##### Bagged-Boosted RENT

The original RENT [17] exploited three statistical characteristics of the covariate coefficients fitted by elastic net for feature selection: (i) the empirical probability of the weights being non-zero; (ii) the rate of the weights having stable signs (positive/negative) across resampled runs; and (iii) the statistical significance obtained using the Student’s *t*-test to reject the null hypothesis of weight = 0. Motivated by the well-established agreement in the literature that ensemble models generally have better representation power than singular models [27], we propose the bagged-boosted RENT (BB-RENT) model, which further improves the feature selection stability of RENT based on the combination of the bagging and boosting ensemble techniques, as summarized below.

First, we propose fitting the boosted elastic net instead of a single elastic net in each of the *K* runs of RENT. This would allow each elastic net in the boosted chain to focus on different portions of the training data, expanding the learnable knowledge domain compared to that without boosting. In this study, we boosted a maximum of *N* elastic nets with boosting coefficients fitted by AdaBoost [28]. Radiomic features with elastic net coefficients weighted by boosting coefficients that matched the abovementioned three criteria [17] after fitting the *K* boosted elastic nets were included.

Second, we further wrapped the boosted RENT with the bagging technique (bootstrap aggregation). In this study, the training data were bootstrapped *Q* times into *Q* subsets, each of which was processed with the boosted RENT, returning *Q* sets of nominated radiomic features. These features were then sieved by considering their nomination frequencies as the inclusion criterion, with features that turned up at a higher frequency suggesting better stability and relevance to the problem of interest. Features with a nomination frequency greater than *η* ⋅ *Q* were ultimately included in the radiomics model-building step.

In this study, we empirically tested different combinations and found the optimal hyperparameters to be *n* = 25, *K* = 200, *Q* = 150, and *η* = 50% for selecting the radiomic features with high stability for further evaluation. Details of the implementation of the proposed BB-RENT are provided in the (Appendix A). An illustration of the proposed method is shown in Figure 3.

### 2.5. Discrimination of NPC and Benign Hyperplasia Using Selected Features

Radiomic models for discriminating early-stage T1 NPC and benign hyperplasia in the T2-fs MRI were built using (a) linear support vector regression (SVR), (b) logistic regression, (c) random forest (RF), (d) perceptron, and (e) *k*-nearest neighbors (*k*NN). Except for LR, which performed binary class predictions, all of these classifiers output the prediction as a continuous number, where a larger value indicates a higher probability of NPC. Z-score normalization was applied to all of the features in order to normalize their magnitude prior to feeding them into the classifiers. The best combinations of the model-fitting hyperparameters (i.e., the parameters that govern the fitting policies of machine learning models) for training each classifier are reported in the Appendix A. We used the relevant implementation in the Python package Scikit-learn for model training and classification [29].

### 2.6. Evaluating Discrimination Performance, Feature Selection Stability, and Statistical Analysis

Next, we evaluated the stability of the radiomic features selected by the proposed BB-RENT and the performance of the proposed pipelines in discriminating early-stage T1 NPC from benign hyperplasia.

#### 2.6.1. Building the Radiomics Model

To investigate the potential of features selected by the proposed BB-RENT in discriminating early-stage T1 NPC from BH in T2-fs MRI, we built a radiomic pipeline comprised of preliminary feature filtering, followed by the proposed BB-RENT and the five corresponding machine learning classifiers (i.e., SVR, logistic regression, RF, perceptron, and *k*NN). This pipeline was first validated with fivefold cross validation, in which the folds were divided among the 3 T cohort with stratification according to the ratio between NPC and BH samples.

The characteristics of the patients in each fold group were evaluated, and potential differences in the patient characteristics across folds were investigated using the Kruskal–Wallis test. Diagnostic accuracy, measured with respect to the area under the receiver operator characteristic curve (AUC), was computed for each fold and each machine learning model used. The mean sensitivity, specificity, and accuracy across folds were computed using thresholds that maximized the Youden index [30]. In addition, the 95% confidence intervals of the performance metrics were estimated by bootstrapping the result 1000 times. The machine learning model with the best performance among the tested classifiers was elected to build the final model.

The final ensemble model was built by aggregating the five classifiers trained corresponding to the fivefold cross validation through their weighted sum. The weights were determined using the reciprocal of their Youden index thresholds for discriminating between NPC and BH, which naturally differed, resulting in normalization of the threshold to ≥1 for NPC in the 3 T MRI training cohort.

#### 2.6.2. Testing the Final Radiomics Model

We tested the performance of the final trained model using another internal patient cohort, that is, the 1.5 T MRI testing cohort. Again, the AUC was evaluated to measure its performance, and the accuracy, sensitivity, and specificity were computed using the normalized threshold of ≥1 for NPC from the 3 T MRI cohort. In addition, the standard Youden index threshold for the 1.5 T MRI testing cohort was also used. The 95% confidence intervals of these results were estimated by bootstrapping 1000 times.

#### 2.6.3. Evaluating the Stability of the Radiomic Features Selected by the Proposed BB-RENT

To understand whether the radiomics features selected by the proposed BB-RENT were stable against changes in available data, we randomly resampled 80% of the patients in the 3 T MRI cohort 100 times without replacement, such that each subset comprised different combinations of patients. For each of these 100 subsets, features were selected using the following five methods: (i) a single elastic net run; (ii) RENT; (iii) boosted RENT; (iv) bagged RENT; and (v) BB-RENT. The features nominated in these 100 runs were recorded, and the stabilities of the five methods were assessed by observing the similarity of the 100 feature sets they nominated. The similarity between the nominated feature sets was measured using the arithmetic mean of the Jaccard index (JAC). As the JAC only measures the similarity between two feature sets, we also evaluated the stability score of Nogueira et al. (NS) [31], which accounts for the similarity between more than two feature sets.

The definitions of mean JAC and NS are reported in the Appendix A. Differences between the mean JAC of the five methods were statistically tested using an independent samples *t*-test, and those of NS were statistically tested using a method similar to the independent samples *t*-test reported by Nogueira et al. [31].

#### 2.6.4. Statistical Analysis

All statistical analyses were performed using either SPSS (IBM SPSS Statistics for Windows, Version 27.0.; IBM Corp., Armonk, NY, USA) or open-source Python packages, including that provided by Nogueira et al. [31]. All experiments were conducted on a machine equipped with two Xeon Gold 5120 (Intel, Santa Clara, CA, USA) processing units with 256 GB of RAM.

## 3. Results

### 3.1. Patient Characteristics

Details of the patient characteristics for the two patient cohorts (i.e., the 3 T MRI training and 1.5 T MRI testing cohorts) and patients in each of the five folds of the 3 T MRI cohort are shown in Table 2. There were no statistical differences in terms of age, sex, or pathology between the two patient cohorts and among different folds (Table 2).

### 3.2. Performance of the Radiomic Model in the 3 T MRI Training Cohort

#### 3.2.1. Fivefold Cross Validation

Using the features selected by the proposed BB-RENT, the performances of the five machine learning models are summarized in Table 3 in terms of discriminating early-stage T1 NPC from BH in the 3 T cohort. Furthermore, the corresponding ROC curve is plotted in Figure 4. The mean AUC for discriminating between NPC and benign hyperplasia across the five folds was 0.85 ± 0.04, 0.77 ± 0.01, 0.82 ± 0.02, 0.84 ± 0.03, and 0.82 ± 0.03 for the models built using SVR, logistic regression, RF, perceptron, and *k*NN, respectively. Each model’s thresholds for identifying NPC were obtained by maximizing the Youden index in order to evaluate their sensitivities and specificities, which are also tabulated in Table 3. Among the five models, SVR showed the best discriminating performance, with AUC, sensitivity, and specificity of 0.85 (95% confidence interval (CI): 0.82–0.89), 79.6% (95% CI: 71.1–88.1%), and 80.8% (95% CI: 72.0–89.6%), respectively, with CI computed based on the fivefold cross validation.

BB-RENT nominated the features “surface volume ratio”, “surface area”, and “long-run high-gray-level emphasis of local binary pattern” in all five folds of the cross validation and the texture features “mean of LoG” and “kurtosis of local binary pattern” in four out of five folds of the cross validation. Further details of the selected features in each fold and the predictions made by each model are provided in Appendix A.

#### 3.2.2. Building the Final Ensemble Radiomic Model

SVR presented the best overall performance during the internal validation; hence, a final model was constructed based on SVR, which included a total of 17 features. The ensemble weights, intercepts, and fitted coefficients are given in Table 4. It should be noted that standardization was performed prior to feeding the features into the ensemble model. The standardization means and variances are reported in Appendix A.

The ensemble model for prediction is therefore:$$\begin{array}{cc} \mathrm{Ensemble}\left(X\right) & =0.33508\cdot \mathrm{SVR}1\left(X^{\left(1\right)}\right)+0.36677\cdot \mathrm{SVR}2\left(X^{\left(2\right)}\right)\\ & +0.62281\cdot \mathrm{SVR}3\left(X^{\left(3\right)}\right)+0.42553\cdot \mathrm{SVR}4\left(X^{\left(4\right)}\right)\\ & +0.33341\cdot \mathrm{SVR}5\left(X^{\left(5\right)}\right), \end{array}$$
where **X** is the Z-score standardized radiomic feature vector, and the superscript **X***^(i)^* indicates the subset of features that was selected for the *i*^th^ fold. The weights of each model were calculated to normalize the Youden index threshold to ≥1 for NPC.

### 3.3. Performance of the Final Model on the 1.5 T Testing Cohort

The ensemble model built using the 3 T MRI training cohort was tested with the 1.5 T MRI testing cohort. This ensemble model obtained an AUC of 0.80 (95% CI: 0.74–0.86). Using the original normalized threshold of ≥1 for NPC, the ensemble model showed an accuracy, sensitivity, and specificity of 66.2% (95% CI: 59.6–72.8%), 87.9% (95% CI: 81.1–93.6%), and 47.4% (95% CI: 37.6–56.3%), respectively. Using the Youden index threshold of ≥1.29 obtained from the 1.5 T MRI testing cohort, the accuracy, sensitivity, and specificity were 74.2% (95% CI: 68.5–80.3%), 76.8% (95% CI: 67.9–84.6%), and 71.9% (95% CI: 63.2–80.0%), respectively.

### 3.4. Stability of Radiomic Feature Selection by Different Feature Selection Methods

Regarding feature selection stability, BB-RENT showed significantly better stability for radiomic feature selection in terms of both NS (all *p* < 0.001) and JAC (all *p* < 0.001) when compared to those selected by the other four methods. The details of the stability scores and statistical analysis results for each of the five feature selection methods are tabulated in Table 5 and plotted in Figure 5.

In agreement with the results from the fivefold cross validation of the performance evaluation, the shape features “surface area” and “surface-to-volume ratio” were consistently selected with the highest frequencies across all five methods. Both features were included in more than 90/100 runs using BB-RENT, where, on average, 7 features were selected from the 422 features. In addition, several texture features, although with slightly lower nomination frequencies and oscillating frequency ranks, were consistently included by all five methods. Those texture features were the “long-run high-gray-level emphasis (LRHLE)” of local kurtosis “(lbp-3D-k, glrlm, LRHLE)” and “mean” of Laplacian of Gaussian (LoG) filtered lesion “(log-sigma-0.4492-mm-3D, first order, mean)”. The interactions between these three features are plotted in Figure 6, and the details of the frequencies of features are reported in the Appendix A.

## 4. Discussion

In this study, we evaluated the role of radiomics analysis in discrimination between early-stage T1 NPC and BH on non-contrast-enhanced T2w-fs MRI and verified that radiomic models can successfully discriminate between the two abnormalities in the nasopharynx. The radiomic model built using stable features selected by the proposed BB-RENT and extracted from patients from the 3 T MRI training cohort achieved an AUC of 0.85 (95% CI: 0.82–0.89) when evaluated using fivefold cross validation for discrimination. A final ensemble model was constructed by aggregating the five models corresponding to the five folds, which involved a total of 17 radiomic features. When this final model was applied to a second internal cohort—the 1.5 T MRI testing cohort—the ensemble model was still able to discriminate between the two abnormalities, albeit with a slightly reduced performance (AUC = 0.80; 95% CI: 0.74–0.86). This close performance was achieved despite the fact that the radiomic model was tested on images obtained at a different Tesla strength (i.e., 1.5 T instead of 3 T) and on two 1.5 T scanners from different manufacturers. The diagnostic performance of this model in terms of accuracy, sensitivity, and specificity was 74.2% (95% CI: 68.5–80.3%), 76.8% (95% CI: 67.9–84.6%), and 71.9% (CI: 63.2–80.0%), respectively, for discriminating early-stage T1 NPC and BH. This radiomic model, using non-contrast-enhanced MRI, could aid NPC screening programs by assessing whether screen-positive patients who have nasopharyngeal lesions are at risk of NPC, thus avoiding unnecessary biopsies in those who turn out to have no cancer.

In the second part of the study, we addressed the stability issue of radiomics, which has posed a key obstacle to its implementation in clinical workflows. In this regard, we proposed a new feature selection algorithm, BB-RENT, based on a combination of the bagging and boosting ensemble techniques for building radiomics models. To discriminate between early NPC and benign hyperplasia in T2w-fs images, we compared the stability of our proposed BB-RENT with well-established feature selection methods using the 3 T MRI cohort. The results showed a significantly higher feature selection stability with BB-RENT when compared to the original algorithm, RENT, and standard elastic net, both when measured in NS (0.54, 0.37, and 0.34, respectively; all *p* < 0.001) and mean JAC (0.39 ± 0.14, 0.25 ± 0.13, and 0.24 ± 0.06, respectively; all *p* < 0.001).

In particular, BB-RENT enabled some degree of interpretation by consistently selecting three features in all folds. Two of these selected features were shape features: “surface area” and “surface–volume ratio”. We postulate that these features—especially surface-to-volume ratio—may reflect the differences in shape observed in the MRI between early-stage T1 NPC (focal mass or diffuse asymmetrical thickening) and BH (diffuse symmetrical thickening) [6]. BB-RENT also selected a texture feature, “lbp-3D-k, glrlm, LRHLE”. A high value of LRHLE in lbp-3D-k suggests that the source image contains small homogenous patches (higher values of local kurtosis among pixels in a neighborhood with ~1 mm radius). Its relevance to the difference between NPC and BH is not as clear as that of shape features, but the negative coefficient consistently fitted by all folds might imply that BH has a higher LRHLE than NPC and therefore more homogeneous patches. Although early CNN studies have suggested that the performances of CNNs in discriminating between early NPC and BH in CT [32] and MRI [8,9] were higher than those reported for radiomics in this study, this ability of radiomics—that is, to identify specific features—provides a major advantage over CNNs in terms of understanding the results of machine learning studies.

In addition, we found that using bagging and boosting alone (i.e., bagged RENT and boosted RENT) did not improve or even lead to a reduction in the feature selection stability (NS = 0.34 and 0.20, respectively) when compared to the original RENT (NS = 0.37). For bagging, this result did not come as a surprise, as RENT already utilizes resampling, which is similar to the principle of bagging. Thus, adding another layer of bagging did not significantly change the performance. For boosting, the decline in stability may be attributed to its well-known susceptibility to noise, as feature selection is an inherently noisy task. Regardless, our results showed that when bagging and boosting were used in combination with RENT (BB-RENT), the stability of the selected radiomic features improved. In the proposed BB-RENT, bagging the boosted feature selection method can overcome the susceptibility to imaging noise associated with boosting. Furthermore, BB-RENT maintained the advantages of RENT in terms of limiting the number of selected features and therefore the risk of overfitting.

The next step in the radiomic pipeline was to discriminate between NPC and BH based on the features selected by BB-RENT to build the final radiomic model. An ensemble model was built by aggregating the five SVRs trained using the 3 T MRI training cohort during fivefold cross validation. This unavoidably resulted in five different thresholds, which needed to be normalized to ≥1 for NPC based on the 3 T MRI training cohort in order to produce a single threshold for practical applications. However, when this normalized threshold was applied to the 1.5 T MRI testing cohort, it no longer represented the point that maximizes the Youden index. In our study, the normalized threshold shifted the discrimination performance in favor of sensitivity in the 1.5 T MRI testing cohort. This minor detail has been frequently overlooked, despite the fact that it may have a profound impact on the sensitivity–specificity tradeoff; as demonstrated in our example, the sensitivity and specificity shifted from 87.9% to 76.8% and from 47.4% to 71.9%, respectively, after calibrating the normalized threshold to maximize the Youden index using the 1.5 T MRI testing cohort. This important issue has been under-reported in the literature. Many radiomics studies have only reported the performance based on the threshold evaluated using the testing set or even failed to report from which set the threshold was derived, thus hindering extrapolation of the reported results to external data. Our results highlight that calibration of the threshold is necessary to obtain the desirable tradeoff for the particular cohort being examined, especially when there is a known shift in data acquisition protocols.

This study is subject to some limitations. First, our method for feature selection is computationally intensive. BB-RENT required 3 h to identify features from the MRIs of 442 patients once, compared to 1 min using RENT; however, this was only required during the training phase and was essential to ensuring the feature selection stability. Once the features are selected and the model is built, BB-RENT require less than a few seconds to make a prediction. Second, this study involved only one institution; however, to evaluate the generalizability of our methods, we included two cohorts, including MRIs obtained from different MRI scanners with two field strengths (3 T and 1.5 T). Although the quality of the images in the 1.5 T cohort did not match that in the 3 T cohort, the model trained using the 3 T cohort was still able to achieve stable performance, only falling slightly behind the internal validation performance. Furthermore, radiomics involves many hyperparameters that may affect the results, including feature extraction settings, feature selection thresholds, and model-training parameters; therefore, an exhaustive search for the best combination of hyperparameters is impractical. In this study, we mainly used the default hyperparameters recommended in the original publications and performed a grid search during the model-building phase. Third, radiomics models have the weakness of requiring the segmentation of lesions, which could limit their value in aiding workflows. Regardless, we previously reported success in using a CNN to segment the primary tumor of NPC in T2w-fs [33,34], which can be expanded to segment any mucosal thickening (including BH) observed in T2w-fs. Combined with this CNN, a fully automatic radiomics method that is useful for clinical workflows can be developed from based on work reported in this study.

## 5. Conclusions

In conclusion, we have proposed an ensemble radiomic model that can discriminate early NPC from BH in non-contrast-enhanced T2w-fs MRIs. Our proposed feature selection method, BB-RENT, achieved improvements in feature selection stability for radiomics analysis. This allowed us to identify consistent features, which can be explained in terms of the known difference in shape between the two entities and identifying texture features that suggest differences in homogeneity. By using the proposed BB-RENT in the feature selection step and SVR—which showed the best performance in classification—we proposed a radiomics model that can discriminate between early-stage T1 NPC and BH in T2w-fs, showing an AUC of 0.85 when cross-validated with the 3 T MRI training cohort and an AUC of 0.80 when tested with the 1.5 T MRI testing cohort. This radiomics model using non-contrast-enhanced MRI may be valuable in automating NPC screening programs for the investigation of individuals with a screen-positive result in an EBV-related blood test for NPC.

## Figures and Tables

**Figure 1 cancers-14-03433-f001:**
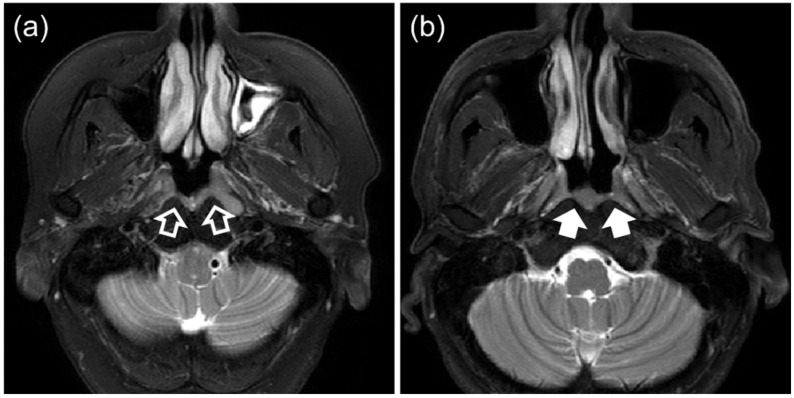
T2-weighted fat-suppressed MRI of patients with (**a**) early-stage T1 nasopharyngeal carcinoma (open arrows) and (**b**) benign hyperplasia (solid arrows).

**Figure 2 cancers-14-03433-f002:**
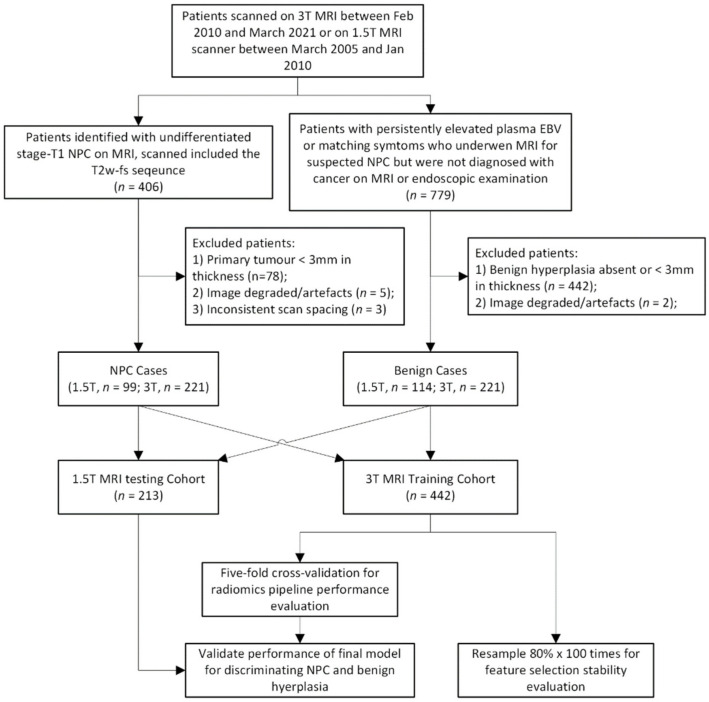
Flow chart showing the study process. NPC = nasopharyngeal carcinoma, MRI = magnetic resonance imaging, EBV = Epstein–Barr virus.

**Figure 3 cancers-14-03433-f003:**
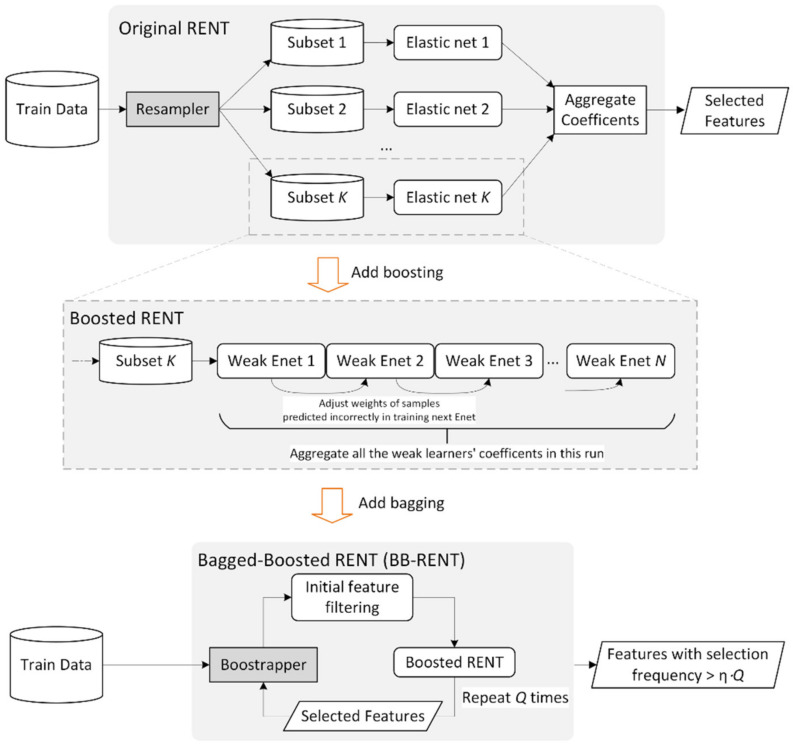
Illustration of the original repeated elastic net technique (RENT), a technique previously reported in [17], and the proposed bagged-boosted RENT (BB-RENT) technique based on it. *K*, *N*, *Q*, and *η* are hyperparameters of the algorithm, where K is the number of times the elastic net or the boosted elastic net is repeated, *N* decides the maximum number of weak learners (depth) used when boosting the elastic nets, *Q* decides the number of times the boosted RENT is bootstrapped and then aggregated, and *η* is the percentage threshold for including a feature in the final selected subset. Enet = elastic net.

**Figure 4 cancers-14-03433-f004:**
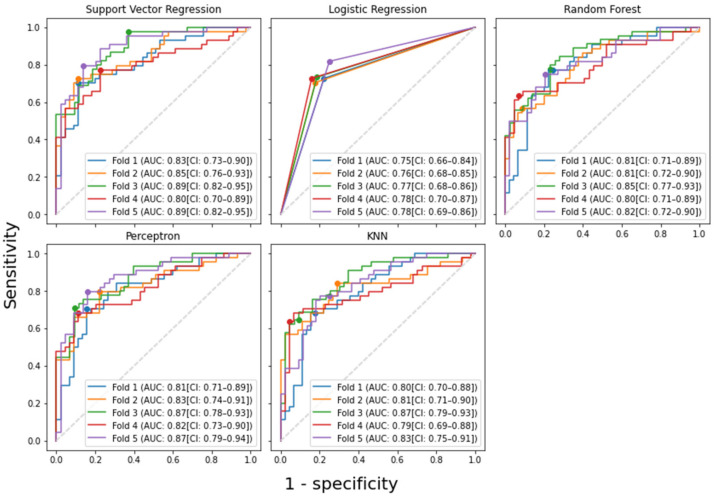
The receiver operator characteristic (ROC) curves of various machine learning models in discriminating stage T1 nasopharyngeal carcinoma (NPC) from benign hyperplasia in fivefold cross validation using the features selected by the proposed-RENT. The 95% confidence interval (CI) of each ROC curve was computed by bootstrapping the data 1000 times. As logistic regression resulted in a binary output of either 0 (for benign hyperplasia) or 1 (for NPC), the ROC curve has only three points in each fold. The dots on the graphs correspond to where the maximum Youden index was achieved. Support vector regression obtained the highest mean AUC (0.850) among the five tested models. *k*NN = *k*-nearest neighbors, AUC = area under the ROC curve, CI = 95% confidence interval.

**Figure 5 cancers-14-03433-f005:**
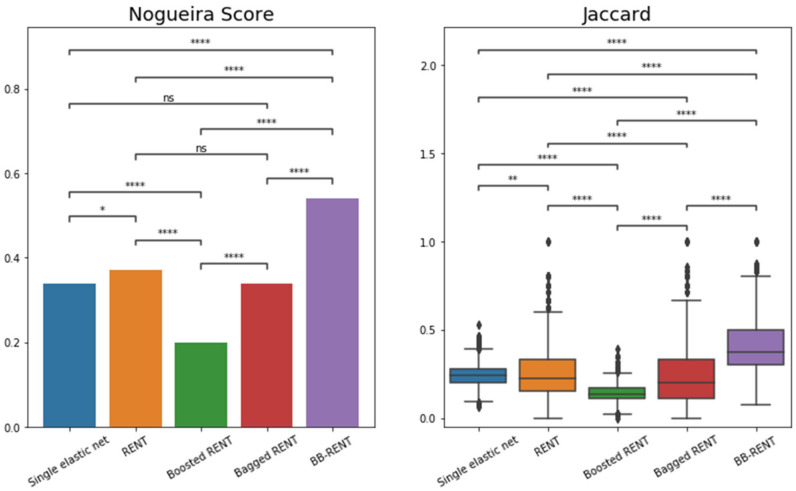
Graph of the feature selection stability results in terms of Nogueira score [31] and Jaccard index for the five feature selection methods. Of the original data, 80% was resampled 100 times, after which the five methods were used for feature selection. The Nogueira scores of each method were compared using the statistical test proposed in [31]. Jaccard indices (JAC) were computed from pairs of features nominated in those 100 runs and compared between different methods using independent sample *t*-tests. RENT = repeated elastic net technique, BB-RENT = bagged-boosted RENT, ns = not significant, * = *p* < 0.05, ** = *p* < 0.01, **** = *p* < 0.0001.

**Figure 6 cancers-14-03433-f006:**
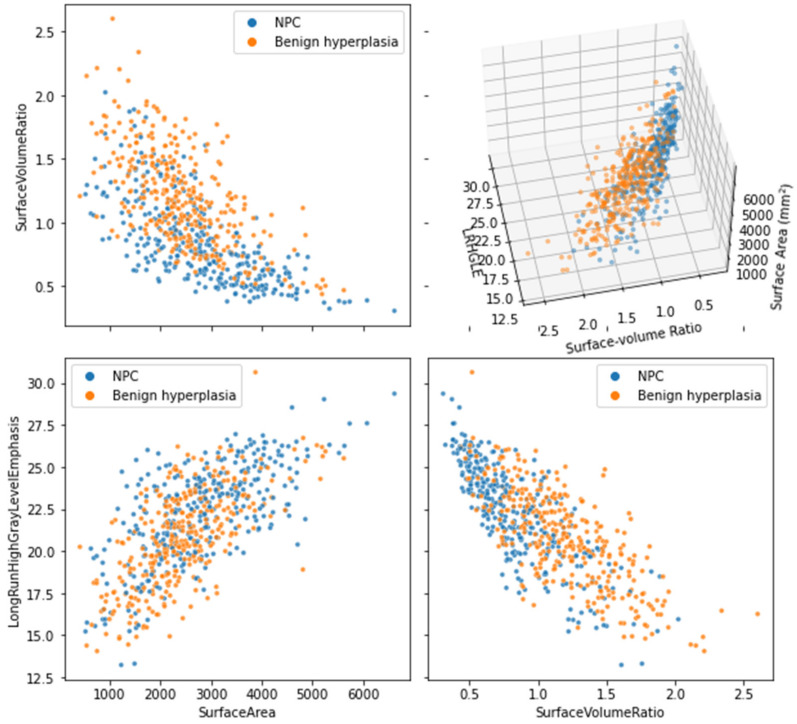
Scatter plots joining the three features: (i) surface–volume ratio, (ii) surface area, and (iii) long-run high-gray-level emphasis of local kurtosis extracted from all patients (i.e., lbp-3D-k, glrlm, LongRunHighGrayLevelEmphasis in PyRadiomics), including both the 3 T training and 1.5 T testing cohorts, which were consistently selected by the proposed BB-RENT method and which were assigned negative coefficients with highest magnitudes across all five folds in the model-training phase. These three features extracted from the T2-weighted, fat-suppressed MRI exhibited potential to discriminate early NPC from benign hyperplasia and were selected by BB-RENT from a total of 422 features. NPC = nasopharyngeal carcinoma, LRHGE = long-run high-gray-level emphasis.

**Table 1 cancers-14-03433-t001:** Summary of previous convolutional neural network studies for the discrimination of nasopharyngeal carcinoma (NPC) and benign hyperplasia on MRI.

Ref.	Year	Cohort	MRI Sequence	AUC	Strengths
Ke et al. [9]	2020	3142 NPC; 958 BH	T1w-ce	0.95	Large cohort; Performed segmentation and differentiation of NPC and BH simultaneously;
Wong et al. [7]	2021	203 stage T1 NPC; 209 BH	T2w-fs	0.96	Study designed to include only stage T1 NPC, which is more challenging than including all T stages to discriminate from BH; Non-contrast-enhanced MRI investigated, which could be used for screening;
Deng et al. [8]	2022	3582 NPC; 825 BH & 71 IM	T1w-ce, T1w, T2w	1.0	Large cohort from the same institution as Ke et al. [9] but also including non-contrast-enhanced MRI; Segmentation and differentiation of NPC and BH performed simultaneously High AUC performance of nearly 1 achieved;

NPC = nasopharyngeal carcinoma, BH = benign hyperplasia, IM = chronic inflammation, T1w-ce = T1-weighted contrast-enhanced, T2w = T2-weighted, T2w-fs = T2w fat-suppressed, AUC = area under the curve.

**Table 2 cancers-14-03433-t002:** Patient characteristics of the 3 T MRI training cohort and the 1.5 T testing cohort, as well as statistical analysis. Patients scanned with a 3 T MRI were sampled randomly with stratification, considering the ratio of nasopharyngeal carcinoma (NPC) to benign hyperplasia (BH), into five folds for later model building and cross validation. Non-parametric ANOVA was used to evaluate differences in characteristics across folds of 3 T MRI patients.

Variable	All 1.5 T (*n* = 213)	All 3 T (*n* = 442)	Fold 1 (*n* = 89)	Fold 2 (*n* = 89)	Fold 3 (*n* = 88)	Fold 4 (*n* = 88)	Fold 5 (*n* = 88)	*p*
Age (years)	48.2 ± 12.7 18–83	54.3 ± 9.5 25–90	54.6 ± 10.2 35–90	53.7 ± 8.2 32–76	55.1 ± 10.1 32–80	52.9 ± 10.2 25–86	55.2 ± 8.34 34–71	0.22
Sex								0.42
Men	139	392	83	77	80	77	75	-
Women	74	50	6	12	8	11	13	-
Pathology								1.00
T1 NPC	99	220	44	44	45	44	44	-
BH	114	222	45	45	43	44	44	-

For age, the data presented denote mean ± standard error (range).

**Table 3 cancers-14-03433-t003:** Performances of various models in discriminating between stage T1 nasopharyngeal carcinoma (NPC) and benign hyperplasia in T2-weighted, fat-suppressed MRI. The accuracy, sensitivity, and specificity were evaluated using the threshold determined by the point maximizing the Youden index in the corresponding fold. As logistic regression returned binary results in our implementation, its threshold is effectively ≥1 for NPC.

Method	AUC	Accuracy	Sensitivity	Specificity	Threshold (≥)
SVR	0.85 ± 0.04	80.3% ± 1.8%	79.6% ± 9.7%	80.8% ± 10.0%	0.51 ± 0.10
Logistic Regression	0.77 ± 0.01	77.2% ± 1.2%	74.2% ± 3.9%	80.1% ± 3.3%	1
RF	0.82 ± 0.02	76.7% ± 1.4%	70.1% ± 8.4%	83.2% ± 7.4%	0.54 ± 0.10
Perceptron	0.84 ± 0.03	79.4% ± 1.6%	73.8% ± 4.8%	85.1% ± 4.4%	0.51 ± 0.06
kNN	0.82 ± 0.03	77.2% ± 1.4%	71.5% ± 7.9%	82.9% ± 9.2%	0.55 ± 0.10

The data presented are the average across the five folds ± standard error. Thresholds indicate the lower inclusive bound for a positive indication of NPC. SVR = support vector regression, RF = random forest, *k*NN = *k*-nearest neighbors.

**Table 4 cancers-14-03433-t004:** Coefficients of the linear support vector regression in the ensemble model proposed to discriminate early nasopharyngeal carcinoma (NPC) from benign hyperplasia on MRI. Features were ordered based on their nomination frequency across the five models, as well as their values in the first fold’s support vector regression (SVR1). These features were selected from a pool of 422 features using the proposed boosted-bagged repeated elastic net technique (BB-RENT). The ensemble weights were determined using the reciprocal of the Youden index threshold divided by the number of models in the ensemble, after which the threshold for nasopharyngeal carcinoma was normalized to ≥1.

Imaging Filter	Feature Type	Feature Name	SVR 1	SVR 2	SVR 3	SVR 4	SVR 5
original	shape	SurfaceVolumeRatio	−0.34350	−0.41914	−0.35446	−0.41196	−0.35104
lbp-3D-k	glrlm	LongRunHighGrayLevelEmphasis	−0.23201	−0.23496	−0.19496	−0.23516	−0.18680
original	shape	SurfaceArea	−0.15209	−0.12242	−0.13933	−0.05791	−0.10701
lbp-3D-m2	first-order	Kurtosis	−0.07727	−0.16856	−0.15919	−0.17252	-
log-sigma−0-4492-mm-3D	first-order	Mean	0.08879	-	0.05899	−0.06722	0.15951
original	shape	LeastAxisLength	0.10404	0.11893	0.10616	-	-
exponential	glcm	SumEntropy	0.04001	-	-	−0.00654	0.13320
lbp-3D-m1	first-order	Kurtosis	−0.08219	-	-	-	−0.14464
gradient	first-order	Energy	0.05771	0.01792	-	-	-
lbp-2D	glcm	DifferenceVariance	−0.02907	-	-	-	-
exponential	first-order	Energy	-	−0.05008	-	-	0.02571
exponential	first-order	Variance	-	-	-	-	−0.12470
exponential	glrlm	RunVariance	-	−0.04025	-	-	-
lbp-3D-m1	glcm	ClusterShade	-	-	−0.06451	-	-
lbp-3D-m2	glrlm	ShortRunHighGrayLevelEmphasis	-	0.00498	-	-	-
log-sigma−0-4492-mm-3D	first-order	RobustMeanAbsoluteDeviation	-	-	-	−0.10530	-
original	glrlm	RunEntropy	-	-	-	-	0.05171
Intercepts			0.48176	0.49719	0.49814	0.47659	0.47752
$\mathrm{Ensemble}\;\mathrm{weight}={\left(threshold_{i}\times 5\right)}^{-1}$			0.33508	0.36677	0.62281	0.42553	0.33341

SVR = support vector regression, *threshold_i_* = Youden index threshold obtained for each fold.

**Table 5 cancers-14-03433-t005:** Feature selection stability in terms of Nogueira score [31] and Jaccard index (JAC) of the five feature selection methods. Whereas boosting and bagging used separately with RENT resulted in worse or similar stability, combing them into BB-RENT led to significantly improved stability. All pairwise comparisons for JAC between the five configurations showed significant differences.

Variable	Single EN	RENT	Boosted RENT	Bagged RENT	BB-RENT
Mean # features (*n* = 100)	30.6 ± 4.0	5.7 ± 1.5	24.9 ± 0.7	5.2 ± 1.6	6.6 ± 1.7
Nogueira score [31]	0.34	0.37	0.20	0.34	0.54
JAC (n=C2100 = 4950)	0.24 ± 0.06	0.25 ± 0.13	0.14 ± 0.05	0.23 ± 0.15	0.39 ± 0.14
***t*-test *p*-values [31]**					
RENT	**0.047**	-	-	-	-
Boosted RENT	**<0.001**	**<0.001**	-	-	-
Bagged RENT	**0.962**	**0.162**	**<0.001**	-	-
BB-RENT	**<0.001**	**<0.001**	**<0.001**	**<0.001**	-

Bold *p*-values indicate significant differences (*p* < 0.05) from the independent *t*-test. RENT = repeated elastic net technique, BB-RENT = bagged-boosted RENT, Mean # features = average number of features nominated across 100 runs, JAC = Jaccard index.

## Data Availability

Data and clinical information presented in this study can be provided upon request to the corresponding authors.

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
