# Peer review of "Radiomics for Discrimination between Early-Stage Nasopharyngeal Carcinoma and Benign Hyperplasia with Stable Feature Selection on MRI"

_cancers, 2022, doi:10.3390/cancers14143433_

Round 1

Reviewer 1 Report

In their manuscript, the authors introduced a new model for differentiating between early-stage nasopharyngeal carcinoma and benign hyperplasia using two datasets. The model introduces a new feature selection method. This is an interesting study. The manuscript is basically well written, but I have some minor comments regarding the manuscript, and I hope the authors will consider addressing them.

1.                  Please include the comparison table after the introduction that should summarize all previous studies and must highlight the strengths of all previous methods. I also suggest citing more relevant and recent literature.

2.                  I am following a group of researchers who work in computer-aided diagnosis systems using radiomics, and I couldn’t find the related work up to date with the recent prospective studies. So, the related work should be updated by adding such 2022 references, for example:

- Ayyad, Sarah M., et al. (2022). " A New Framework for Precise Identification of Prostatic Adenocarcinoma." Sensors, 22(5), 1848.

- Zeng, F., et al. (2022). MRI-based radiomics models can improve prognosis prediction for nasopharyngeal carcinoma with neoadjuvant chemotherapy. Magnetic Resonance Imaging, 88, 108-115.

3.                  Is the model available online in some way?

4.                  There is a typo in line 368 “Error! Reference source not found.”, please remove it.

5.                  Please add another experiment without using feature selection, “using the whole feature”

Author Response

We would like to thank the reviewer for their efforts in reviewing our manuscript and their valuable comments. We have response to the comments point-by-point. Please see the attachment.

Reviewer 2 Report

Very interesting, well designed and conducted work on important topic of early detection of nasopharyngeal carcinoma. Authors presented their own radiomic model of dicrimination of the early nasopharyngeal carcinoma from benign hiperplasia of the nasopharyngeal tissue in non-contrast enhanced MRI. This would be very important diagnostic and screening tool.

When assessing the manuscript from a purely clinical point of view, the greatest value of the conducted studies is the achievement of a diagnostic model that allows for the identification of patients with lesions in the nasopharynx suggesting a malignant neoplasm, in whom unnecessary invasive diagnostic procedures can be avoided. The diagnostic performance of this model in terms of accuracy, sensitivity, and specificity, was 74.2% (95% CI: 68.5–80.3%), 76.8% (95% CI: 67.9–84.6%), and 71.9% (CI: 63.2–80.0%), respectively, for discriminating early stage-T1 NPC and BH.

Authors proofed the stability of proposed radiomics model, the BB-RENT based on bagging and boosting ensemble. The results showed a significantly higher feature selection stability with BB- RENT, when compared to the original algorithm RENT and standard elastic net in the stability score (NS) as well as arithmetic mean of the Jaccard index (JAC).

The strength of this study is a concept of the relatively stable and reliable radiomic model to differentiate between NPC and BH, that although still not ready to use, might be an option for further investigations and development.

I can’t see any essential weak points of this study, some existing limitations of this work are mentioned by the authors in Discussion paragraph.

Author Response

We would like to deliver our gratitude for agreeing to review our manuscript and thank you very much for your valuable comments.